# TOPOLOGICAL POSITIONAL ENCODING

## ABSTRACT

Unlike words in sentences, nodes in general graphs do not have canonical positional information. As a result, the local message-passing framework of popular graph neural networks (GNNs) fails to leverage possibly relevant global structures for the task at hand. In this context, positional encoding methods emerge as an efficient approach to enrich the representational power of GNNs, helping them break node symmetries in input graphs. Similarly, multiscale topological descriptors based on persistent homology have also been integrated into GNNs to boost their expressivity. However, it remains unclear how positional encoding interplays with PH-based topological features and whether we can align the two to improve expressivity further. We address this issue with a novel notion of topological positional encoding (ToPE) that amalgamates the strengths of persistence homology and positional encoding. We establish that ToPE has provable expressivity benefits. Strong empirical assessments further underscore the effectiveness of the proposed method on several graph and language processing applications, including molecular property prediction, out-of-distribution generalization, and synthetic tree tasks.

## 1 INTRODUCTION

Many natural systems, such as social networks (Freeman, 2004) and proteins (Jha et al., 2022), exhibit complex relational structures often represented as graphs. To tackle prediction problems in these domains, graph neural networks (GNNs) (Scarselli et al., 2009; Bronstein et al., 2017; Hamilton et al., 2017; Velickovic et al., 2017) have become the dominant approach, leading to breakthroughs in diverse applications such as drug discovery (Gilmer et al., 2017b; Stokes et al., 2020; Satorras et al., 2021), simulation of physical systems (Cranmer et al., 2019; Sanchez-Gonzalez et al., 2020), algorithmic reasoning (Dudzik et al., 2023; Jurss et al., 2023), and recommender systems (Ying et al., 2018). Despite this success, most GNNs have rather limited expressivity — they are at most as powerful as the 1-Weisfeiler-Lehman (1-WL) test (Weisfeiler & Leman, 1968) in distinguishing non-isomorphic graphs (Xu et al., 2019; Morris et al., 2019). This inherent limitation has prompted the development of more expressive GNNs by leveraging, e.g., topological features (Horn et al., 2022), random features (Sato et al., 2021), higher-order message passing (Morris et al., 2019), and structural/positional encodings (Li et al., 2020; You et al., 2019; Wang et al., 2022).

Inspired by the success of positional encodings (PEs) in Transformers (Vaswani et al., 2017) for sequences, several positional encoders for graphs have been proposed (You et al., 2019; Dwivedi et al., 2021; Wang et al., 2022; Huang et al., 2023). For instance, spectral methods exploit global structure via the eigendecomposition of the graph Laplacian (Lim et al., 2022; Kreuzer et al., 2021; Huang et al., 2023). However, these encodings suffer from inherent ambiguities due to sign flips, basis changes, stability, and eigenvalue multiplicities. Recent efforts have addressed sign and basis symmetries (Lim et al., 2022; Wang et al., 2022) and stability with respect to graph perturbations (Huang et al., 2023). However, a common drawback persists: most methods partition the Laplacian eigenvalue/eigenvector space and utilize only the partitioned eigenvalues/eigenvectors. This approach discards valuable information contained in the remaining eigenvalues and eigenvectors. Another class of methods leverage relative distances (e.g., computed from random walk diffusion) to anchor-nodes to capture structural information (Dwivedi et al., 2021; Eliasof et al., 2023; Ying et al., 2021; You et al., 2019; Li et al., 2020). Despite these advances, existing methods fail to extract detailed multiscale topological information, such as the persistence of connected components and independent cycles (i.e., 0- and 1-dim topological invariants), which may be relevant to downstream tasks and potentially more expressive.

Persistent homology (PH) (Edelsbrunner et al., 2002) is the cornerstone of topological data analysis and offers a powerful framework to capture multi-scale topological information from data. In the context of graphs, PH has been recently used, e.g., to boost the expressive and representational power of GNNs (Horn et al., 2022; Immonen et al., 2023; Carriere et al., 2020; Verma et al., 2024). However, integrating PH-based topological descriptors into graph positional encoders remains unexplored.

In this work, we explore persistent homology on graphs to build expressive positional encoders. In particular, we present Topological Position Encoding (ToPE) — a multilayered encoding scheme that builds on top of prior positional encoders and the message-passing paradigm (Gilmer et al., 2017a) to obtain filtering functions used to compute persistent topological features (i.e., persistence diagrams). The resulting node-level topological embeddings are fed to downstream GNN layers. Notably, ToPE is very flexible as it can be combined with any PE method and addresses the limitations of current methods by leveraging more nuanced, fine-grained topological information captured by PH.

We theoretically analyze ToPE and establish its improved expressive power in comparison to popular PE methods such as the Laplacian PE (Dwivedi et al., 2021). We also show the stability of ToPE's filtration functions and describe its representational power in terms of the $k$-WL hierarchy. To demonstrate the effectiveness of our proposal, we conduct rigorous empirical evaluations on various tasks, including molecule property prediction, out-of-distribution generalization, and synthetic tree tasks.

In sum, our main contributions are:

1. (**Methodology**) We propose Topological Positional Encoding (ToPE), unifying general positional encoding schemes with persistent homology on graphs;

2. (**Theory**) We establish a series of theoretical results to support our methodological contribution. In particular, we demonstrate the superior expressive power of ToPE compared to its base positional encoders, and ToPE's relationship with the $k$-WL isomorphism test;

3. (**Empirical**) Our empirical assessment shows that ToPE outperforms the competing baselines across diverse tasks such as molecular property prediction, out-of-distribution generalization tasks, and synthetic tree tasks. We also conduct an ablation study to measure the impact of adopting learnable vs. non-learnable filtering functions.

## 2 BACKGROUND

In this section, we overview prior graph positional encoding methods, the $k$-dim Weisfeiler-Leman test, and some basic notions in persistent homology for graph data.

**Notation.** We define a graph as a tuple $G = (V, E, x)$, where $V = \{1, \ldots, n\}$ is the set of nodes, $E \subseteq V \times V$ is the set of edges, and the function $x : V \to \mathbb{R}^{d_x}$ assigns a color (or $d_x$-dimensional feature vector) to nodes $v \in V$ — for convenience, hereafter, we denote the feature vector of $v$ by $x_v$. We denote the adjacency matrix of $G$ by $A \in \{0, 1\}^{n \times n}$, i.e., $A_{ij}$ is one if $(i, j) \in E$ and zero otherwise. We use $D$ to represent the diagonal degree matrix of $G$, i.e., $D_{ii} = \sum_j A_{ij}$. We define the normalized Laplacian of $G$ as $\Delta = I_n - D^{-1/2}AD^{-1/2}$ and its random walk Laplacian as $\Delta_{\mathrm{RW}} = D^{-1}A$, where $I_n$ is the $n$-dimensional identity matrix. The set of neighbors of a node $v$ is denoted by $\mathcal{N}(v) = \{u \in V : (v, u) \in E\}$. Furthermore, we use $\{\!\{\cdot\}\!\}$ to denote multisets.

### 2.1 GRAPH POSITIONAL ENCODING

Given a graph $G$, a positional encoder acts on $A$ (adjacency matrix of $G$) to obtain an embedding matrix $P \in \mathbb{R}^{n \times k}$, where the $v$-th row of $P$ comprise the positional feature of node $v$, denoted by $p_v$. Integrating PEs into message-passing GNNs (Gilmer et al., 2017a; Xu et al., 2019) enables them to learn intricate relationships between nodes based on positional information, ultimately enhancing their representational power. Although several PE methods (Dwivedi et al., 2021; Li et al., 2020; Lim et al., 2022; Wang et al., 2022; Bo et al., 2023) have been proposed, most approaches build upon:

- Laplacian PE (Dwivedi & Bresson, 2020): This approach employs the idea of Laplacian eigenmaps (Belkin & Niyogi, 2003) as PE. In particular, let $\Delta = U\Lambda U^\top$, where $U \in \mathbb{R}^{n \times n}$ is an orthonormal matrix with eigenvectors $u_1, \ldots, u_n$ and the matrix $\Lambda = \mathrm{diag}(\lambda_1, \ldots, \lambda_n)$ comprises the corresponding eigenvalues (or spectrum) of $\Delta$, with $\lambda_1 \leq \lambda_2 \leq \cdots \leq \lambda_n$.

Then, Laplacian PE uses the $k$ smallest (non-trivial) eigenvectors as positional encodings, i.e., $p_v = [u_{1,v}, u_{2,v}, \ldots, u_{k,v}]$ for all $v \in V$. We note that this corresponds to the solution to: $\max_{P \in \mathbb{R}^{n \times k}} \mathrm{trace}(P^\top \Delta P)$ subject to $P^\top DP = I_k$.

- Distance PE (Li et al., 2020): Let $S \subseteq V$ be a target subset of vertices. Distance PE learns node features for each node $v$ based on distances from $v$ to elements in $S$ (You et al., 2019). The distances comprise either random walk probabilities or generalized PageRank scores (Pal et al., 2017; Mialon et al., 2021). Formally, using sum-pooling, Distance PE computes $p_v = \sum_{s \in S} f(d_G(v, s))$ with $d_G(v, s) = [(\Delta_{\mathrm{RW}})_{vs}, (\Delta^2_{\mathrm{RW}})_{vs}, \ldots, (\Delta^k_{\mathrm{RW}})_{vs}]$ or $d_G(v, s) = (\sum_{i=1}^{k} \gamma_i \Delta^i_{\mathrm{RW}})_{vs}$, where $\gamma_i \in \mathbb{R}$ and $f(\cdot)$ is a multilayer perceptron.

- Random walk PE (Dwivedi et al., 2021): This approach captures node proximity through the random walk diffusion process and can be viewed as a simplified version of Distance PE. In particular, Dwivedi et al. (2021) adopt $p_v = [(\Delta_{\mathrm{RW}})_{vv}, (\Delta^2_{\mathrm{RW}})_{vv}, \ldots, (\Delta^k_{\mathrm{RW}})_{vv}]$.

Dwivedi et al. (2021) also propose *learnable structural and positional encodings* (LSPE) as a general framework that builds upon base positional encoders (e.g., LapPE). More specifically, the key idea of LPSE lies at decoupling positional and structural representations and learn them using message-passing layers. Formally, starting from $x_v^0 = x_v$ and $p_v^0 = p_v \; \forall v \in V$, LSPE recursively updates positional and node embeddings as

$$x_v^{\ell+1} = \mathrm{Upd}_\ell^x \left(x_v^\ell, p_v^\ell, \mathrm{Agg}_\ell^x(\{\!\{x_u^\ell, p_u^\ell : u \in \mathcal{N}(v)\}\!\})\right), \quad \forall v \in V,$$

$$p_v^{\ell+1} = \mathrm{Upd}_\ell^p \left(p_v^\ell, \mathrm{Agg}_\ell^p(\{\!\{p_u^\ell : u \in \mathcal{N}(v)\}\!\})\right), \quad \forall v \in V,$$

where $\mathrm{Agg}_\ell^p$ and $\mathrm{Agg}_\ell^x$ are arbitrary order-invariant functions, and $\mathrm{Upd}_\ell^x$ and $\mathrm{Upd}_\ell^p$ are arbitrary functions (often multilayer perceptrons, MLPs). After iterative updating, the final layer node embeddings are concatenated with the final positional ones, i.e., $\{[x_v^L, p_v^L]\}_v$, and then leveraged for downstream tasks, such as node classification, graph classification, or link prediction.

## 2.2 $k$-DIM WEISFEILER-LEMAN TEST

The Weisfeiler–Leman test (1-WL), also known as the color refinement algorithm (Weisfeiler & Leman, 1968), aims to determine if two graphs are isomorphic. It does so by iteratively assigning colors to nodes. Initially, nodes receive labels based on their features. In each iteration, nodes sharing the same label get distinct labels if their sets of similarly labeled neighbors differ. Termination happens when label counts diverge between graphs, indicating non-isomorphism.

Due to the shortcomings of the 1-WL in distinguishing non-isomorphic graphs, Babai (1979); Immerman & Lander (1990) introduced a more powerful variant known as $k$-dim (*folklore*) Weisfeiler–Leman algorithm. In this approach, $k$-FWL colors subgraphs instead of a single node. Specifically, given a graph $G$, it colors tuples from $V(G)^k$ for $k \geq 1$ instead of nodes and defines neighborhoods between these tuples. Formally, let $G$ be a graph, and let $k \geq 2$. If $\mathbf{v} \in V(G)^k$, then $G[\mathbf{v}]$ is the subgraph induced by the components of $\mathbf{v}$, where the nodes are labeled with integers from $\{1, ..., k\}$ corresponding to indices of $\mathbf{v}$.

In each iteration $i \geq 0$, the algorithm computes a *coloring* $C_i^k : V(G)^k \to \mathbb{N}$, and in the initial iteration ($i = 0$) two tuples $\mathbf{v}$ and $\mathbf{w}$ in $V(G)^k$ get the same color if the map $v_i \to w_i$ induces an isomorphism between $G[\mathbf{v}]$ and $G[\mathbf{w}]$. For $i > 0$, the algorithm proceeds as,

$$C_{i+1}^k(\mathbf{v}) = \mathrm{RELABEL}\left((C_i^k(\mathbf{v}), M(\mathbf{v}))\right) \tag{1}$$

where the multi-set $M(\mathbf{v}) = \{\!\{C_i^k(\phi_1(\mathbf{v}, w)), \ldots, C_i^k(\phi_k(\mathbf{v}, w)) \mid w \in V(G)\}\!\}$ and $\phi_j(\mathbf{v}, w) = \{v_1, \ldots, v_{j-1}, w, v_{j+1}, w_k\}$. The $\phi_j(\mathbf{v}, w)$ replaces the $j$-th component of the tuple $\mathbf{v}$ with the node $w$. Consequently, two tuples are adjacent or $j$-neighbors (with respect to a node $w$) if they differ in the $j$th component (or are equal, in the case of self-loops). The algorithm iterates until convergence, i.e., $C_i^k(\mathbf{v}) = C_i^k(\mathbf{w}) \iff C_{i+1}^k(\mathbf{v}) = C_{i+1}^k(\mathbf{w})$ for all $\mathbf{v}$, defining the stable partition induced by $C_i^k$, define $C_\infty^k(\mathbf{v}) = C_i^k(\mathbf{v})$. The algorithm then proceeds analogously to the 1-WL.

We say that the $k$-FWL distinguishes two graphs $G$ and $H$ if their color histograms differ. This means there exist a color $c$ in the image of $C_\infty^k$ such that $G$ and $H$ have distinct numbers of node tuples of color $c$. Morris et al. (2023) also describe another variant of $k$-WL known as $k$-dim (*oblivious*) WL algorithm. The key distinction between the two lies in aggregating over different neighborhoods. In

this case, for each position $j \in [k]$ we obtain a set of $|V(G)|$ neighbors by replacing $v_j$ by $w \in V$. A hash for position $j$ is obtained using these colors, and the overall color is obtained by aggregating the hashed colors across all $k$ positions (and $v$'s color from previous iteration). We utilize the former variant throughout the paper and refer to Morris et al. (2023); Huang & Villar (2021) for a thorough discussion of the algorithm and its properties.

## 2.3 PERSISTENT HOMOLOGY ON GRAPHS

A key notion in persistent homology is that of filtration. In this regard, a *filtration* of a graph $G$ is a finite nested sequence of subgraphs of $G$, i.e., $\emptyset = G_0 \subset G_1 \subset ... \subset G$. A popular choice to obtain a filtration consists of considering sublevel sets of a function defined on the vertices of a graph. In particular, let $f : V \to \mathbb{R}$ be a filtering function and $G_\alpha$ be the subraph of $G$ induced by the vertex set $V_\alpha = \{v : f(v) \leq \alpha\}$ for $\alpha \in \mathbb{R}$. By varying $\alpha$ from $-\infty$ to $\infty$, we obtain a sub-level filtration of $G$. Importantly, we can monitor the emergence and vanishing of topological characteristics (e.g., connected components, loops, voids) throughout a filtration, which is the core idea of PH. More specifically, if a topological feature first appears in $G_{\alpha_b}$ and disappears in $G_{\alpha_d}$, then we encode its persistence as a pair $(\alpha_b, \alpha_d)$; if a feature does not disappear, then its persistence is $(\alpha_b, \infty)$. The collection of all pairs forms a multiset that we call *persistence diagram*. We use $\mathcal{D}^i$ to denote the persistence diagram for $i$-dim topological features. For a formal treatment of PH, we refer to Edelsbrunner & Harer (2010) and Hensel et al. (2021).

In graph learning, persistent homology has been harnessed to enhance the expressive power of GNNs. Horn et al. (2022) introduced TOGL, a general framework for integrating topological features derived from PH into GNN layers. TOGL employs a learnable function (a multilayer perceptron, MLP) on node features / colors to obtain graph filtrations, which we refer to as *vertex-color* (VC) filtrations. Importantly, Immonen et al. (2023) characterized the expressive power of VC filtrations via the notions of *color-separating sets* and *component-wise colors*. Formally, a color-separating set for a pair of graphs $G = (V, E, x)$ and $G' = (V', E', x')$ is a set of colors $Q$ such that the subgraphs induced by $V \setminus \{w \in V \mid x_w \in Q\}$ and $V' \setminus \{w \in V' \mid x'_w \in Q\}$ have distinct component-wise colors — defined as the multiset comprising the set of node colors of each connected components. Immonen et al. (2023) introduced RePHINE, a new topological descriptor that employs vertex and edge-color filtering functions to enhance the expressiveness of persistence diagrams from VC filtrations.

# 3 TOPOLOGICAL POSITIONAL ENCODING

In this section, we introduce *topological positional encodings* (ToPE, in short). ToPE builds on top of existing graph PE methods, leveraging them to obtain detailed topological information of graphs via persistent homology. Here, we also analyze stability and expressivity properties of our proposal.

## 3.1 METHOD

**PE-based message passing.** Let $p_v \in \mathbb{R}^d$ be a base PE (e.g., Laplacian PE) for a node $v \in V(G)$. We first propagate positional embeddings over the graph following a vanilla message-passing procedure. In particular, starting from $p_v^0 = p_v$ for all $v$, we recursively update the positional embeddings as

$$p_v^{\ell+1} = \text{Upd}_\ell^p \left( p_v^\ell, \text{Agg}_\ell^p(\{\!\{ p_u^\ell : u \in \mathcal{N}(v) \}\!\}) \right) \quad \forall v \in V, \tag{2}$$

where $\text{Agg}_\ell^p$ is an order-invariant function and $\text{Upd}_\ell^p$ is an arbitrary update function at layer $\ell$. Similarly to LSPE, ToPE decouples positional and feature propagation steps.

**Using PEs to induce graph filtrations.** The second step in ToPE consists of using the positional encodings $\{p_v^\ell\}_v$ as node features to compute vertex-color (or edge-color) filtrations (Horn et al., 2022; Immonen et al., 2023). More specifically, we define the filtration function at layer $\ell$ as the map $p_v^\ell \mapsto f^\ell(p_v^\ell) \in \mathbb{R}$ to obtain a sub-level filtration induced by $f^\ell$, i.e., given the graph $G = (V, E, x)$, we build $G_\alpha = (V_\alpha, E_\alpha, x)$ at filtration step $\alpha$ by setting $V_\alpha = \{v \in V : f^\ell(p_v^\ell) \leq \alpha\}$ and $E_\alpha = \{(u, v) \in E : \max\{f^\ell(p_u^\ell), f^\ell(p_v^\ell)\} \leq \alpha\}$. For simplicity, we have adopted only one filtration function per layer, although multiple ones can be considered. From the filtration $\{G_\alpha\}_\alpha$, we compute its 0-dim persistence diagram at layer $\ell$, denoted by $\mathcal{D}_\ell^0$.

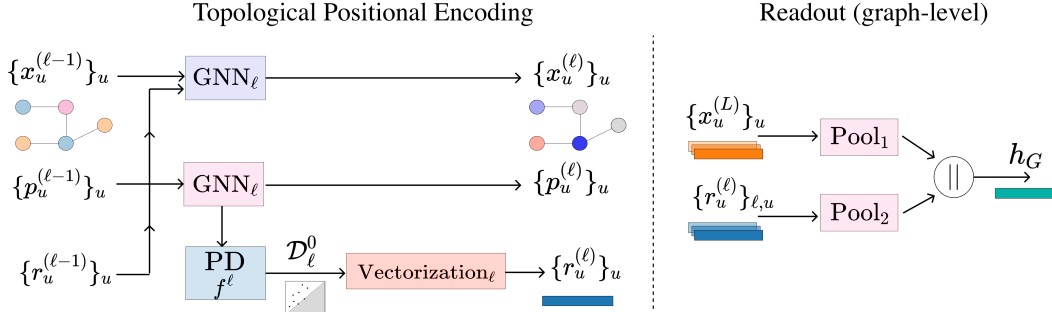

Figure 1: **Overview of ToPE.** At each layer $\ell$, the node embeddings $\{x_u^{\ell-1}\}_u$ are updated using the positional embeddings $\{p_u^{\ell-1}\}_u$ and the topological embeddings $\{r_u^{\ell-1}\}_u$ as described in Eq. 3. The position embeddings $\{p_u^{\ell-1}\}_u$ are updated and then leading to the computation of persistance diagrams $\mathcal{D}^0$ leading to topological embeddings $\{r_u^\ell\}_u$. In readout phase, the final layer node embeddings $\{x_u^L\}_u$ are combined with the topological embeddings $\{r_u^\ell\}_{u,\ell}$ for various tasks.

We note that our design also accommodate other descriptors. For instance, one may also compute RePHINE diagrams (Immonen et al., 2023) or extended persistence diagrams (Carriere et al., 2020). The key idea here consists of using base positional features to obtain filtering functions.

**Obtaining topological embeddings.** Importantly, we can associate persistence pairs in $\mathcal{D}_\ell^0$ with nodes in $V(G)$ — there is a bijection from $\mathcal{D}_\ell^0$ to $V$, which was also explored in (Horn et al., 2022). Let $t_v^\ell$ be the tuple in $\mathcal{D}_\ell^0$ associated with node $v$. Then, we vectorize each element $t_v^\ell$ using an MLP $\phi_\ell$ to obtain the embeddings $r_v^\ell = \phi_\ell(t_v^\ell)$ for all $v \in V(G)$. We refer to $r_v^\ell$ as the *topological embedding* associated with the base PE $p_v^\ell$.

**Integrating PEs and topological embeddings into GNNs.** A simple strategy to integrate PEs and topological embeddings into GNNs is to combine (e.g., concatenate or add) them with GNN node embeddings $\{x_v^\ell\}_v$. Then, the resulting GNN's message-passing procedure at layer $\ell$ becomes

$$x_v^{\ell+1} = \text{Upd}_\ell^x\left([x_v^\ell \parallel p_v^\ell \parallel r_v^\ell], \text{Agg}_\ell(\{\!\{[x_u^\ell \parallel p_u^\ell \parallel r_u^\ell] : u \in \mathcal{N}(v)\}\!\})\right) \quad \forall v \in V. \tag{3}$$

**Achieving class predictions.** For graph classification, as usual, we apply a readout function (e.g., sum or mean) to the embeddings at the last GNN layer, $L$, to obtain a graph-level embedding $x_G$, i.e., $x_G = \text{Readout}(\{x_v^L\}_v)$. Similarly to LSPE, we can also concatenate positional embeddings $p_v^L$ with node representations $x_v^L$ before applying the readout function. Then, we combine $x_G$ with a global topological embedding $\text{Pool}(\{r_v^\ell\}_{\ell,v})$ and send the resulting vector through an MLP to achieve graph-level predictions — Pool is either a global mean or addition operator.

Figure 1 describes the architectural steps of our method. Importantly, our framework is versatile and can accommodate any selection of base (initial) positional encoding as well as various topological descriptors (e.g., RePHINE) and message-passing GNNs.

### 3.2 ANALYSIS

We now report results on the stability and expressiviness of leveraging persistent homology to obtain topological positional encodings. All proofs can be found in Appendix B.

Put simply, stability of a method describes that a slight perturbation in the input induces only a minor change in the output. Thus, in a stable positional encoding method, a small perturbation in the input graph should correspond to a small flutuation in the positional encoding. This notion is formalized in Definition 1, following Wang et al. (2022); Huang et al. (2023).

**Definition 1** (Stable PE). *Let $G$ and $G'$ be two graphs with $n$ nodes and corresponding Laplacians $\Delta$ and $\Delta'$. A PE method $\Psi_{\text{PE}}$ is stable if there exists $c, L_\Psi > 0$, such that*

$$||\Psi_{\text{PE}}(\Delta) - P_*\Psi_{\text{PE}}(\Delta')||_F \leq L_\Psi ||\Delta - P_*\Delta' P_*^\top||_F^c, \tag{4}$$

*where $P_* = \arg\min_{P \in \Pi(n)} ||\Delta - P\Delta' P^\top||_F^c$ and $\Pi(n)$ is the set of the $n$-by-$n$ permutation matrices.*

Our next result (Proposition 1) states the stability of the filtering functions used in ToPE composed to base PE methods.

**Proposition 1.** *Let $\Delta$ and $\Delta'$ be two graph Laplacians. If $\Psi_{\mathrm{PE}}$ is stable, then there exist constants $c, L_f > 0$ such that:*

$$||f(\Psi_{\mathrm{PE}}(\Delta)) - P_* f(\Psi_{\mathrm{PE}}(\Delta'))||_F \leq L_f ||\Delta - P_* \Delta' P_*^\top||_F^c \tag{5}$$

*where $P_* = \arg\min_{P \in \Pi(n)} ||\Delta - P \Delta' P^\top||_F^c$, $\Pi(n)$ is the set of the $n$-by-$n$ permutation matrices, and $f$ is the filtration function used by ToPE to compute topological features.*

We now establish theoretical results on the expressive power of building blocks used in ToPE. Lemma 1 shows that defining filtering functions on positional encodings results in 0-dim persistence diagrams that are at least as expressive as the positional encodings in distinguishing non-isomorphic graphs. In other words, we do not lose expressive power by relying only on 0-dim diagrams.

**Lemma 1.** *Let $G = (V, E, \cdot)$ and $G' = (V', E', \cdot)$ be two graphs with associated positional encodings $Z = \{\!\!\{p_v\}\!\!\}_{v \in V}$ and $Z' = \{\!\!\{p'_v\}\!\!\}_{v \in V'}$ obtained from any base PE method. If $Z \neq Z'$, then there exists a vertex-color filtration on the attributed graphs $\tilde{G} = (V, E, p)$ and $\tilde{G}' = (V', E', p')$ such that $\mathcal{D}^0(\tilde{G}) \neq \mathcal{D}^0(\tilde{G}')$.*

In Proposition 2, we show that PH-based topological encodings (i.e., ToPE) are strictly more expressive than a concrete and popular PE method — Laplacian PEs.

**Proposition 2.** *Consider base Laplacian PE positional encodings relying on a fixed number of smallest eigenvalue/eigenvector pairs of graph Laplacians. There are pairs of graphs that Laplacian PE cannot distinguish but ToPE can. The converse does not hold.*

The expressive power of 0-dim diagrams from vertex-color filtrations is fully characterized by the notion of color-separating sets, as outlined by Immonen et al. (2023). Since ToPE combines GNNs with PH, we also provide results on the connection between color-separating sets and the $k$-WL hierarchy. Proposition 3 shows that whenever $k$-FWL distinguishes two graphs, there exists a filtration that produces 0-dim persistence diagrams for these graphs, or equivalently, there is a color-separating set. We also provide an explicit coloring for the graphs based on the stable colorings from $k$-FWL.

**Proposition 3.** *If $k$-FWL deems two graphs $G = (V, E, x)$ and $G' = (V', E', x')$ non-isomorphic with stable colorings $C_\infty : V^k \to \mathbb{N}$ and $C'_\infty : V'^k \to \mathbb{N}$, then $Q = \emptyset$ is a trivial color-separating set for the associated graphs $\tilde{G} = (V, E, \tilde{x})$ and $\tilde{G}' = (V', E', \tilde{x}')$, with $\tilde{x}(u) = \mathrm{hash}(\{C_\infty(v) : u \in v, v \in V^k\})$ for all $u \in V$ and $\tilde{x}'(u) = \mathrm{hash}(\{C'_\infty(v) : u \in v, v \in V'^k\})$ for all $u \in V'$.*

We note that Proposition 3 strengthens the results by Rieck (2023) (Proposition 5) in two ways. First, we show how to harness $k$-FWL to find an specific filtering function that leads to separable diagrams — in fact, given the proposed coloring, separability holds for any injective vertex-color function. Also, with our scheme, even 0-dim diagrams are different, while Proposition 5 in (Rieck, 2023) provides that $k - 1$ or $k$-dim diagrams are different.

## 4 EXPERIMENTS

We assess the performance of ToPE on diverse and challenging tasks. In Section 4.1, we assess its effectiveness in predicting properties of drug molecules and performing real-world graph classification. Section 4.2 delves into ToPE's robustness by benchmarking its ability to handle domain shifts in data. Finally, Section 4.3 showcases the generalizability of ToPE by evaluating its performance on synthetic tree-structured tasks.

**Implementation.** ToPE is implemented in PyTorch (Paszke et al., 2019) and utilise the same training configuration as the competing baselines. More details in Appendix C.

**Baselines.** To empirically demonstrate the effectiveness of our method, we compared it against existing positional encoding approaches on various tasks. We utilized several established baselines for graph tasks: (i) No positional encodings, (ii) SignNet (Lim et al., 2022), (iii) PEG (Wang et al., 2022), (iv) LapPE & RWPE (Dwivedi et al., 2021), and (v) SPE (Huang et al., 2023). In order to

Table 1: **AUC-ROC results**. DrugOOD Benchmark and baselines are taken from Huang et al. (2023). ToPE outperforms the competing baselines in achieving better scores for OOD-Test.

| Domain | PE method | Diagram | ID-Val ↑ | ID-Test↑ | OOD-Val ↑ | OOD-Test↑ |
|---|---|---|---|---|---|---|
| Assay | - | - | $92.92_{\pm 0.14}$ | $92.89_{\pm 0.14}$ | $71.02_{\pm 0.79}$ | $71.68_{\pm 1.10}$ |
| | PEG | - | $92.51_{\pm 0.17}$ | $92.57_{\pm 0.22}$ | $70.86_{\pm 0.44}$ | $71.98_{\pm 0.65}$ |
| | | VC | $92.62_{\pm 0.19}$ | $92.75_{\pm 0.49}$ | $71.62_{\pm 0.57}$ | $72.13_{\pm 0.93}$ |
| | | RePHINE | $92.42_{\pm 0.27}$ | $92.35_{\pm 0.19}$ | $72.02_{\pm 0.51}$ | $\mathbf{72.33}_{\pm 1.03}$ |
| | SignNet | - | $92.26_{\pm 0.21}$ | $92.43_{\pm 0.27}$ | $70.16_{\pm 0.56}$ | $72.27_{\pm 0.97}$ |
| | | VC | $91.66_{\pm 0.39}$ | $92.73_{\pm 0.28}$ | $70.37_{\pm 0.69}$ | $73.07_{\pm 1.07}$ |
| | | RePHINE | $91.36_{\pm 0.31}$ | $92.15_{\pm 0.29}$ | $69.47_{\pm 0.43}$ | $\mathbf{73.87}_{\pm 1.32}$ |
| | SPE | - | $92.84_{\pm 0.20}$ | $92.94_{\pm 0.15}$ | $71.26_{\pm 0.62}$ | $72.53_{\pm 0.66}$ |
| | | VC | $92.78_{\pm 0.96}$ | $92.49_{\pm 0.58}$ | $71.78_{\pm 0.64}$ | $72.91_{\pm 1.16}$ |
| | | RePHINE | $92.16_{\pm 0.37}$ | $93.12_{\pm 0.91}$ | $72.33_{\pm 0.93}$ | $\mathbf{73.11}_{\pm 1.07}$ |
| Scaffold | - | - | $96.56_{\pm 0.10}$ | $87.95_{\pm 0.20}$ | $79.07_{\pm 0.97}$ | $68.00_{\pm 0.60}$ |
| | PEG | - | $95.65_{\pm 0.29}$ | $86.20_{\pm 0.14}$ | $79.17_{\pm 0.29}$ | $69.15_{\pm 0.75}$ |
| | | VC | $96.65_{\pm 0.31}$ | $86.44_{\pm 0.81}$ | $79.79_{\pm 0.47}$ | $\mathbf{70.12}_{\pm 0.52}$ |
| | | RePHINE | $96.94_{\pm 0.62}$ | $86.54_{\pm 0.77}$ | $79.40_{\pm 0.35}$ | $69.31_{\pm 0.97}$ |
| | SignNet | - | $95.48_{\pm 0.34}$ | $86.73_{\pm 0.56}$ | $77.81_{\pm 0.70}$ | $66.43_{\pm 1.06}$ |
| | | VC | $93.03_{\pm 0.57}$ | $83.65_{\pm 0.77}$ | $74.73_{\pm 0.65}$ | $67.37_{\pm 1.12}$ |
| | | RePHINE | $93.35_{\pm 0.56}$ | $85.05_{\pm 0.79}$ | $75.05_{\pm 1.04}$ | $\mathbf{68.03}_{\pm 1.34}$ |
| | SPE | - | $96.32_{\pm 0.28}$ | $88.12_{\pm 0.41}$ | $80.03_{\pm 0.58}$ | $69.64_{\pm 0.49}$ |
| | | VC | $96.57_{\pm 0.43}$ | $88.37_{\pm 0.82}$ | $80.56_{\pm 0.65}$ | $\mathbf{70.92}_{\pm 0.92}$ |
| | | RePHINE | $96.87_{\pm 0.76}$ | $89.98_{\pm 1.05}$ | $80.76_{\pm 0.87}$ | $70.46_{\pm 0.79}$ |

compute the topological descriptors via persistence homology, we utilized (i) Vertex Color (VC) (Horn et al., 2022) and (ii) RePHINE (Immonen et al., 2023) as learnable methods to compute the diagrams. For the synthetic tree tasks, we compared our method against these positional encoding approaches: (i) Sinusodial (Gehring et al., 2017), (ii) Relative (Shaw et al., 2018)n and (iii) RoPE (Su et al., 2024) positional embedding methods.

## 4.1 DRUG MOLECULE PROPERTY PREDICTION AND GRAPH CLASSIFICATION

We used ZINC (Dwivedi et al., 2023) and Alchemy (Chen et al., 2019) datasets which comprises of quantum mechanical properties of drug molecules, with the main goal of predicting these properties. We followed the data preparation strategy of Huang et al. (2023) and utilised the same GIN as base model with training, validation and test splits. For graph classification tasks, we employed two datasets: OGBG-MOLTOX21 (Huang et al., 2017; Wu et al., 2018) and OGBG-MOLPCBA (Wang et al., 2012; Wu et al., 2018). OGBG-MOLTOX21 is a multi-task binary classification dataset containing 7.8k graphs, where the goal is to predict toxicity across 12 measurements for each molecule. OGBG-MOLPCBA is a larger dataset with 437.9k graphs, focusing on predicting activity/inactivity labels for 128 bioassays. To ensure consistent comparisons, we followed the data preparation steps outlined by Dwivedi et al. (2021) and used Gated-GCN as the base architecture.

**Evaluation Results.** Figure 2 presents the test Mean Absolute Error (MAE) between our model's predictions and the ground truth values for the ZINC and Alchemy datasets (property prediction tasks). It also showcases the results on the OGBG-MOLTOX21 and OGBG-MOLPCBA datasets (graph classification tasks). Notably, our method consistently outperforms competing baselines, particularly on the ZINC and MOLPCBA datasets, demonstrating significant improvements. Furthermore, incorporating ToPE with the PEG baseline consistently leads to the largest decrease in MAE across ZINC. This highlights the effectiveness of our approach in capturing richer and more informative graph representations.

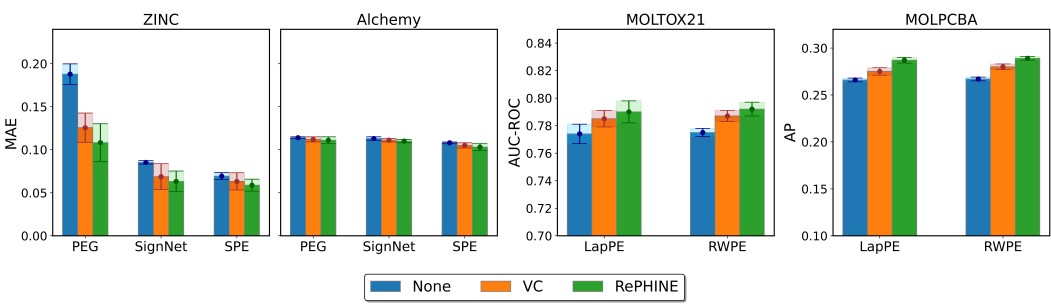

Figure 2: **Test MAE and graph classification results**. Integration of ToPE into general position encoding schemes leads to better downstream performance across diverse datasets.

## 4.2 OUT OF DISTRIBUTION PREDICTION

To evaluate our method's ability to handle domain shifts, we employed the DrugOOD, an out-of-distribution (OOD) benchmark (Ji et al., 2023). We focused on the ligand-based affinity prediction task to assess drug activity. DrugOOD introduces two types of distribution shifts: (i) Assay, denoting the assay to which the data point belongs, and (ii) Scaffold, representing the core structure of molecules. The DrugOOD dataset is divided into five parts: training set, in-distribution (ID) validation/test sets, and out-of-distribution (OOD) validation/test sets. The OOD sets have different underlying distributions compared to the ID sets, allowing us to assess generalizability to unseen data.

Table 2: **Synthetic Tree tasks**. Perplexity (PPL) ↓ on various synthetic tree tasks.

| Scheme | Diagram | $C_3$ | | Reorder | | Copy | |
|---|---|---|---|---|---|---|---|
| | | breadth | depth | breadth | depth | breadth | depth |
| Sinusodial | None | 2.47 | 2.90 | 6.93 | **7.11** | 1.14 | 5.76 |
| | VC | 2.42 | 2.75 | 6.80 | 7.21 | 1.10 | 5.47 |
| | RePHINE | **2.33** | **2.64** | **6.75** | 7.51 | **1.00** | **5.32** |
| Relative | None | 1.85 | 2.62 | 6.00 | **7.72** | 1.10 | 5.94 |
| | VC | **1.53** | 2.42 | **5.92** | 8.11 | 1.01 | 5.04 |
| | RePHINE | 1.70 | **2.31** | 6.12 | 7.97 | **1.00** | **4.82** |
| RoPE | None | 1.84 | 2.52 | 4.93 | 6.63 | 1.85 | 3.17 |
| | VC | 1.65 | 1.94 | 4.76 | 5.24 | 1.14 | 2.35 |
| | RePHINE | **1.59** | **1.77** | **4.49** | **4.70** | **1.00** | **2.05** |

**Superior OOD Generalizability.** Table 1 summarizes the AUC-ROC scores for different methods. Interestingly, all models achieve similar performance on the in-distribution test set (ID-Test). However, performance drops for all methods on the out-of-distribution test set (OOD-Test). This highlights the challenge of generalizing to unseen data. Notably, our method exhibits the best performance on the OOD-Test set. This demonstrates the effectiveness of our approach in capturing features relevant for generalizability, even when encountering unseen data distributions.

## 4.3 SYNTHETIC TREE TASKS

We explored three synthetic tree-tasks involving binary branching trees: (i) Tree-copy, analogous to a sequence copy-task; (ii) Tree-rotation, where the output tree mirrors the input, interchanging left and right children; and (iii) Algebraic expression reduction, where input trees represent complex expressions from the cyclic group $C_3$, and the model is tasked with performing a single reduction step, i.e., reducing all depth-1 subtrees into leaves. We followed the data-preparation strategy of Kogkalidis et al. (2023) and utilized same splits and hyperparameters.

**Improved Performance on Tree Tasks.** Table 2 presents the Perplexity (PPL) scores for all methods on the synthetic tree tasks. Our method consistently achieves lower PPL scores compared to

Table 3: **Identity filtrations (left) and Runtime Comparisons (right)**.

| PE method | Diagram | ZINC |
|---|---|---|
| PEG | - | 0.1878 $\pm 0.012$ |
| | VC-I | 0.1432 $\pm 0.023$ |
| | VC | 0.1256 $\pm 0.017$ |
| | RePHINE | 0.1082 $\pm 0.022$ |
| SPE | - | 0.0693 $\pm 0.004$ |
| | VC-I | 0.0608 $\pm 0.013$ |
| | VC | 0.0599 $\pm 0.010$ |
| | RePHINE | 0.0588 $\pm 0.007$ |

| PE method | Diagram | Alchemy |
|---|---|---|
| PEG | - | 5.90 $\pm 0.40$ |
| | VC | 6.45 $\pm 1.20$ |
| | RePHINE | 6.51 $\pm 0.90$ |
| SPE | - | 16.70 $\pm 1.40$ |
| | VC | 18.70 $\pm 1.70$ |
| | RePHINE | 19.10 $\pm 1.10$ |

the baseline across all tasks. This indicates that incorporating our approach on top of a positional encoding method leads to improved performance on downstream tasks. This finding highlights the versatility of our method, demonstrating its effectiveness across various data domains, including those involving tree-structured data.

## 5 ABLATIONS

**Identity Filtrations.** We investigated the impact of learnable versus non-learnable filtrations in vertex-color (VC) PH method. The persistence diagrams capture the evolving topological features of the data as it is continuously simplified. We compared using positional encodings directly via an identity filtration function (VC-I), to define filtration values for computing persistence diagrams, against a learned filtration function. Table 3 showcases the results alongside comparisons with learnable variants (VC & RePHINE) on ZINC dataset. We observe that using the positional encodings as filtration values to compute the persistence diagrams improves the performance. This is further enhanced by learning a parameterized filtration function, highlighting the method's increased expressiveness.

**Runtime Comparison.** We conducted an ablation study to investigate the computational cost of our method. We measured the time (in seconds) per epoch to train different models on a single V100 GPU. The results for various persistent homology and positional encoding methods are shown in Table 3 over the Alchemy dataset. We observe that SPE introduces additional computational overhead due to its more intensive computations compared to the simpler positional encoding such as PEG. However, adding topological positional encoding (ToPE) on top of these methods does not significantly increase computational complexity, demonstrating the efficiency of our method.

## 6 CONCLUSION, BROADER IMPACT AND LIMITATIONS

We introduce Topological Positional Encoding (ToPE), a novel method that unifies the power of persistent homology (PH) with general positional encoding methods. ToPE leverages rich topological information to enhance the expressivity of GNNs by combining the strengths of persistence homology and positional encoding. We theoretically establish ToPE's improved expressive power compared to Laplacian positional encoding and characterize its representational capabilities within the $k$-WL hierarchy. Our extensive empirical evaluations across diverse tasks demonstrate ToPE's effectiveness, showing significant improvements over existing methods.

While ToPE offers substantial advantages, there are limitations to address. The computational cost associated with persistent homology computations remains an area for exploration. Additionally, this work is currently restricted to graphs that are 1-dim simplicial complexes, limiting its applicability to higher-order structural relationships. Extending ToPE to handle general higher-order complexes presents a promising direction for future research, potentially enabling the capture of more sophisticated topological features. In conclusion, we believe our work paves the road for many advancements in boosting the representational power of GNNs. By establishing a bridge between topological data analysis and positional encodings, ToPE opens new possibilities for developing more effective and expressive graph learning architectures.

REPRODUCIBILITY STATEMENT

The Appendix C provides extensive detail about the dataset used, the method's parameterization, and training details. We have utilized open source datasets and libraries for implementation.

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

## A    RELATED WORKS

**Graph positional encodings.**    Positional encodings enhance representations in Graph Neural Networks (GNNs) (Gilmer et al., 2017a; Xu et al., 2019; Velickovic et al., 2017) by incorporating relational information between nodes based on their positions.  Several approaches have been developed to achieve this, including Laplacian-based methods that utilize the graph laplacian (Dwivedi et al., 2021; Kreuzer et al., 2021; Maskey et al., 2022; Lim et al., 2022; Wang et al., 2022; Huang et al., 2023), random walk-based techniques that leverage walks on graphs (Dwivedi et al., 2021; Eliasof et al., 2023), and PageRank-inspired approaches that compute auxiliary distances (Ying et al., 2021; Li et al., 2020).  However, these methods partition the Laplacian eigenvalue/eigenvector space and utilize only the partitioned eigenvalues/eigenvectors, disregarding the valuable information contained in the remaining eigenvalues and eigenvectors.  To address this limitation, we propose complementing the existing approaches with topological descriptors based on persistent homology, which can capture additional structural information from the graph.

**Persistent homology on graphs**    Persistence homology methods (Horn et al., 2022; Carriere et al., 2020; Immonen et al., 2023) from topological data analysis have made rapid strides, providing topological descriptors that augment GNNs (Cesa & Behboodi, 2023; Verma et al., 2024) with persistent information to obtain more powerful representations, enhancing the expressivity (Rieck, 2023; Wang et al., 2024) and generalizability (Brilliantov et al., 2024).  However, these methods have not been analyzed in regards with positional encodings in graphs, and the unification of these topological descriptors with positional encodings remains an unexplored frontier.

## B    PROOFS

### B.1    PROOF OF PROPOSITION 1

The proof follows from the Lipschitz continuity of the filtering function $f(\cdot)$, which we parametrize using an MLP and, therefore, is Lipschitz continuous. We know from Definition 1 that

$$||\Psi_{\text{PE}}(\Delta) - P_*\Psi_{\text{PE}}(\Delta')||_F^c \leq L_\Psi \cdot ||\Delta - P_*\Delta' P_*^\top||_F^c. \tag{6}$$

We note that $f(\cdot)$ is a row-wise function. Thus, we have that $f(P_*\Psi_{\text{PE}}(\Delta')) = P_*f(\Psi_{\text{PE}}(\Delta'))$. Then, it follows that

$$||f(\Psi_{\text{PE}}(\Delta)) - P_*f(\Psi_{\text{PE}}(\Delta'))||_F^c \leq L_m||\Psi_{\text{PE}}(\Delta) - P_*\Psi_{\text{PE}}(\Delta')||_F^c \tag{7}$$

$$\leq L_m L_\Psi \cdot ||\Delta - P_*\Delta' P_*^\top||_F^c \tag{8}$$

where the last inequality comes from the stability of the base positional encoder $\Psi_{\text{PE}}$.

### B.2    PROOF OF PROPOSITION 2

To prove the Proposition 2, it suffices to i) show a pair of graphs that have same $n$ smallest eigenvalue and eigenvector pairs, ii) show that persistence diagrams for those two graphs are different.

Let $K_i$ denote the complete graph with $i$ nodes. Consider a graph $G = \cup_{i=1}^{n/2} K_1 \cup K_3$ — here $K_1 \cup K_3$ denotes a graph with two components comprising one isolated node and a triangle. Also, consider $G' = \cup_{i=1}^{n/2}(K_1 \cup K_1 \cup K_1 \cup K_1)$ — i.e., $4n/2$ isolated nodes. The $n$ smallest eigenvalues corresponding to $G$ are all equal to 0 with the identical constant eigenvector. Similarly, $G'$ has the same eigenvalues with identical constant eigenvectors. Therefore, Laplacian PE relying on fixed $n$ smallest eigenvalue/eigenvector pairs, cannot distinguish these graphs.

By leveraging Theorem 1 from Immonen et al. (2023), since the number of connected components in $G$ and $G'$ are different, they necessarily have different 0-dimensional persistence diagrams i.e., $\mathcal{D}^0(G) \neq \mathcal{D}^0(G')$ for any color-filtration function over vertices. This difference in persistence diagrams allows us to distinguish between the graphs despite their identical $n$ smallest eigenvalues and eigenvectors.

### B.3 PROOF OF LEMMA 1

To prove the Lemma 1, it suffices to show that the persistence diagram pairs obtained when using $Z, Z'$ as vertex colors are different.

Consider $G = (V, E, \cdot)$ and $G' = (V', E', \cdot)$ be two graphs with associated positional encodings $Z = \{\!\!\{p_v\}\!\!\}_{v \in V}$ and $Z' = \{\!\!\{p'_v\}\!\!\}_{v \in V'}$ obtained from any base PE method. We utilize the positional encodings as vertex colors (or filtration values), on the attributed graphs $\tilde{G} = (V, E, p)$ and $\tilde{G}' = (V', E', p')$ to obtain 0-dim persistence diagrams.

If the positional encodings of $\tilde{G}$ and $\tilde{G}'$ are different, i.e., $Z \neq Z'$, this corresponds to having distinct multisets of vertex colors for the graphs. Hence, by leveraging Lemma 5 in Immonen et al. (2023), the multisets of birth times would be different for persistence diagrams, leading to $\mathcal{D}^0(\tilde{G}) \neq \mathcal{D}^0(\tilde{G}')$.

### B.4 PROOF OF PROPOSITION 3

Consider two graphs $G = (V, E, x)$ and $G' = (V', E', x')$ that are deemed non-isomorphic by $k$-FWL with stable colorings, $C_\infty : V^k \to \mathbb{N}$ and $C'_\infty : V'^k \to \mathbb{N}$.

Then we can use hash functions $\tilde{x}(u) = \text{hash}(\{C_\infty(v) : u \in v, v \in V^k\})$ for all $u \in V$ and $\tilde{x}'(u) = \text{hash}(\{C'_\infty(v) : u \in v, v \in V'^k\})$ for all $u \in V'$, to project the colorings from $k$-tuple of nodes to obtain node colors in the associated set of graphs $\tilde{G} = (V, E, \tilde{x})$ and $\tilde{G}' = (V', E', \tilde{x}')$. Since, hash functions are injective in nature, and $G$ and $G'$ can be distinguished via $k$-FWL, then there exists color of a tuple $\mathbf{v}_w \in V^k$ such that $C_\infty(\mathbf{v}_w) \neq C'_\infty(\mathbf{v}'_w), \forall \mathbf{v}'_w \in V'^k$. Then, any node in $v \in \mathbf{v}_w$ (note that $v \in V$) will have a color that is not in nodes of $V'$, i.e., $\tilde{x}_v \neq \tilde{x}'_u$ for all $u \in V'$. This will provide us with different node colors for the associated graphs $\tilde{G}$ and $\tilde{G}'$. Hence, by leveraging the definition of color-separating sets from Immonen et al. (2023), $Q = \phi$ is a trivial color-separating set for graphs $\tilde{G}$ and $\tilde{G}'$.

## C IMPLEMENTATION DETAILS

Below are the implementation details. We trained all the methods on a single NVIDIA V100 GPU.

### C.1 DRUG MOLECULE PROPERTY PREDICTION AND GRAPH CLASSIFICATION

We adhered to the precise hyperparameters and training configuration outlined in Huang et al. (2023) for predicting drug molecule properties and in Dwivedi et al. (2021) for classifying real-world graphs in our experiments. For graph classification, we used Gated-GCN (Kipf & Welling, 2017) as our base model. To compute the Persistence Homology (PH) diagrams, we employed the learnable PH method proposed by Immonen et al. (2023). The PH layers operated exclusively on the position encoding features of every layer with the following specified hyperparameters in Table 4.

Table 4: Default hyperparameters for RePHINE/VC method

| Hyperparameter | Meaning | Value |
|---|---|---|
| PH embed dim | Latent dimension of PH features | 64 |
| Num Filt | Number of filtrations | 8 |
| Hiden Filtration | Hidden dimension of filtration functions | 16 |

### C.2 OUT OF DISTRIBUTION PREDICTION

We adhered to the precise hyperparameters and training configuration outlined in Huang et al. (2023) for Drug OOD benchmark. To compute the Persistence Homology (PH) diagrams, we employed the learnable PH method proposed by Immonen et al. (2023). The PH layers operated exclusively on the position encoding features of every layer with the following specified hyperparameters in Table 4.

## C.3 SYNTHETIC TREE TASKS

We created the synthetic tree dataset by sampling random trees of maximum depths from a discretized normal $\mathcal{N}(7,1)$ and followed similar training setup as described in Kogkalidis et al. (2023). We adhered to the hyper-parameters and training configuration used in Kogkalidis et al. (2023) and employed the PH-layers on top of the position encoding features with an additional layer to update position encodings, using hyper-parameters described in Table 5.

Table 5: Default hyperparameters for RePHINE/VC method

| Hyperparameter | Meaning | Value |
|---|---|---|
| PH embed dim | Latent dimension of PH features | 64 |
| Num Filt | Number of filtrations | 8 |
| Hiden Filtration | Hidden dimension of filtration functions | 128 |

