# OpenReview forum: "Topological Positional Encoding"
_ICLR.cc/2025/Conference — Submitted to ICLR 2025_

### Official Review · Reviewer_Dztj · 2024-11-03

**Soundness:** 2
**Presentation:** 2
**Contribution:** 2
**Rating:** 5
**Confidence:** 3

**Summary:**

The authors proposes Topological Positional Encodings (ToPE), which leverages persistence diagram of filtrations constructed by Laplacian positional encodings. They show the superior expressive power of ToPE over vanilla Laplacian positional encodings as well as a connection to k-FWL test.  Experiments on molecular graphs and synthetic tree tasks show promising performance of ToPE.

**Strengths:**

1. a novel positional encodings that encodes topological features using learnable persistence diagrams
2. the paper is easy to follow and well-structured

**Weaknesses:**

One thing is unclear to me is the motivation to build filtraction upon positional encodings rather than other node features. What is the key role of positional encodings here and is there any intuition behind? For example, if we suppose the positional encodings here only serve for a better distinguishability of nodes, then what if we define filtraction on node features based on high-order GNNs and even random node features. Will it lead to a even much more expressive model?

**Questions:**

Given that persistence diagrams are discrete function of graph filtrations, how are we supposed to train it using gradient descent?

---

> ### Author Response · Authors · 2024-11-24
> **Response**
>
> Thank you very much for your feedback! We address your questions and comments below.
>
> > One thing is unclear to me is the motivation to build filtraction upon positional encodings....
>
> Thanks for the opportunity to clarify this. The role of positional encodings is to capture global positional information --- i.e., information that only depends on the graph structure and not the features. The same idea has been used, e.g., in Transformers, to encode sequences. Many papers have shown the gains/importance of PE for graphs [1,2,3].
>
> The rationale of our approach is that existing positional encodings (even when combined with GNNs) fail to capture information that PH can capture. Thus, we propose a new positional encoding method that builds on top of any prior PE, improving its expressivity.
>
> Regarding the use of high-order GNNs and random node features, you are right, we can use them to increase the expressive power of PH-based graph models. However, higher-order GNNs are often costly, and random features break the equivariance of GNNs.
>
> [1] Haorui Wang,et. al, Equivariant and stable positional encoding for more powerful graph neural networks, ICLR 2022.
>
> [2] Derek Lim,  et. al, Sign and basis invariant networks for spectral graph representation learning, ICLR 2023
>
> [3] Vijay Prakash Dwivedi, et. al, Graph neural networks with learnable structural and positional representations, ICLR 2022
>
>
> > Given that persistence diagrams are discrete function of graph filtrations, how are we supposed to train it using gradient descent?
>
> Thanks for your question. When learning PH-based topological features, we apply a filtration function $f_\theta$ (e.g., an MLP) and a learnable vectorization of the tuples of the diagrams. Note that the tuples contain the filter values obtained by $f_\theta$ and we only need differentiability wrt $\theta$. In [1], the authors establish conditions for differentiability of persistence computation (see Lemma 1, [1]). In particular, they show that if $f$ is injective at node level (i.e., $f(v) \neq f(v')$ if $v \neq v'$), the gradients are well-defined. If not, the computation of gradients may depend on specific implementations --- the same issue can happen when we have set operations (e.g., maxpooling) in neural networks. For the sake of analysis, authors in [2] (Lemma 1) show we can relax that requirement since it is  always possible to build an injective filtration function that is arbitrarily close to a non-injective one.
>
> [1] Graph Filtration Learning. ICML, 2020.
>
> [2] Topological Graph Neural Networks. ICLR, 2022.

---

### Official Review · Reviewer_G43i · 2024-11-03

**Soundness:** 2
**Presentation:** 3
**Contribution:** 2
**Rating:** 5
**Confidence:** 4

**Summary:**

The paper introduces Topological Positional Encoding (ToPE), a new technique that combines persistent homology (PH) with existing positional encoding strategies to augment the capabilities of Graph Neural Networks (GNNs). ToPE theoretically improved expressive power compared to Laplacian positional encoding and shows strong generalization capabilities.

**Strengths:**

-I have found the theory well written and self-contained. I think a non-expert could find most of the information in the paper, and I appreciate this aspect.

-The insights are didactical and well communicated. The conclusion given by the experiments looks interesting and valuable for future practitioners, while I think a synthesis would be beneficial for the reader.

**Weaknesses:**

Despite these merits, I have the following concerns about the paper.

1- While there is a careful analysis of the different design decisions/performance tradeoffs, I feel that there is only a limited understanding about what are the properties of the Architecture that lead to these decisions/performance differences.

2-  The paper does not include an analysis of the hyperparameters for the proposed approach, even though previous work, such as 'Where Did the Gap Go? [Tönshoff 2023],' has demonstrated that these hyperparameters can significantly influence the performance of transformer architectures. I recommend that the authors conduct such an analysis to better understand the behavior and performance implications of their architecture.

3- The paper does not sufficiently address the computational cost associated with persistent homology computations, which is known to be substantial. This oversight could limit the practical applicability of the Topological Positional Encoding (ToPE) method, especially in scenarios where computational resources are constrained. Additionally, the paper falls short in comparing ToPE with a wide range of baseline methods, which could have provided a more comprehensive evaluation of its performance and effectiveness relative to existing solutions.

**Questions:**

- Are there specific theoretical or computational challenges you foresee in expanding the applicability of ToPE to capture more complex topological features in such structures?

---

> ### Author Response · Authors · 2024-11-24
> **Response**
>
> Thank you very much for your feedback! We address all your questions and comments below.
>
> > While there is a careful analysis of the different design decisions/performance tradeoffs
>
> The rationale of our approach is that existing positional encodings (even when combined with GNNs) fail to capture information that PH can capture. Thus, we propose a new positional encoding method that builds on top of any prior PE, improving its expressivity.
>
> We have theoretically shown that incorporating topological descriptors boosts the representational power of base PE methods (**Proposition 2, Lemma 1**) and shown that ToPE is also strictly more expressive than PEs combined with GNNs.  Motivated by it, we have defined ToPE, which utilize PH based topological descriptors with GNNs, to obtain a better donwstream performance. Moreover, we also performed an ablation study on ZINC dataset for property prediction (test MAE shown in the table below), where we capture the boost in downstream performance, when utilizing different PH topological descriptors such as vertex-color filtrations or RePHINE.
>
> | PE method | Diagram | ZINC|
> | --------| ------| ------|
> | SPE| - | 0.0693 $\pm$ 0.004|
> | SPE| VC | 0.0599 $\pm$ 0.010 |
> | SPE| RePHINE | 0.0588 $\pm$ 0.007 |
>
> > The paper does not sufficiently address the computational cost ....
>
> **Computational Complexity** We have performed an ablation study to evaluate the computational complexity of adding PH onto the positional encodings for the Alchemy dataset. The table below showcases the time (in seconds) per epoch to train different models on a single V100 GPU. We observe that adding topological positional encoding (ToPE) on top of these methods does not significantly increase computational complexity, demonstrating the efficiency of our method.
>
> | PE method | Diagram | Alchemy|
> | --------| ------| ------|
> | SPE| - | 16.70  $\pm$ 1.40|
> | SPE| VC |18.70  $\pm$ 1.70 |
> | SPE| RePHINE | 19.10 $\pm$ 1.10 |
> | PEG| - |5.90  $\pm$ 0.40|
> | PEG| VC |  6.45  $\pm$ 1.20 |
> | PEG| RePHINE |  6.51 $\pm$ 0.90 |
>
> We have proposed a positional encoding method, and to ensure a fair comparison and demonstrate the benefit of our method, we have adhered to the same data preparation strategy and training/validation/testing splits and compared our method to latest graph positional encoding methods such as SPE[1], PEG[2], SignNet[3], LapPE and RWPE[4].
>
> [1] Yinan Huang, et. al, On the stability of expressive positional encodings for graph neural networks, ICLR 2024.
>
> [2] Haorui Wang,et. al, Equivariant and stable positional encoding for more powerful graph neural networks, ICLR 2022.
>
> [3] Derek Lim,  et. al, Sign and basis invariant networks for spectral graph representation learning, ICLR 2023
>
> [4] Vijay Prakash Dwivedi, et. al, Graph neural networks with learnable structural and positional representations, ICLR 2022
>
> > Are there specific theoretical or computational challenges you foresee in expanding the applicability of ToPE to capture more complex topological features in such structures?
>
> Thanks for your question. Extending ToPE to higher-order simplicial complexes can enhance ToPE to capture complex topological features, however, this comes with certain challenges such as defining PH for higher-order simplicial complexes and defining effective lifting procedures to define higher-order simplicial complexes [1] remains an open area of research.
>
>
> [1] Bernárdez, Guillermo, et al. "ICML Topological Deep Learning Challenge 2024: Beyond the Graph Domain." arXiv preprint arXiv:2409.05211 (2024).

---

### Official Review · Reviewer_72Z5 · 2024-11-03

**Soundness:** 1
**Presentation:** 3
**Contribution:** 1
**Rating:** 1
**Confidence:** 3

**Summary:**

The paper proposes to employ a Topological Positional Encoding (ToPE) during the message passing of a GNN. In practice, during the message passing, three vectors are attached to each node at each layer: the first is the node embedding, the second represents the positional encoding, and the third one is the topological embeddings based on persistent Homology (PH).

The node embeddings are updated by concatenating all three vectors for each node and by applying the common update function of a GNN.

The positional encodings are updated by applying the update function of a GNN (as in standard methods that employ positional encoding).

The topological embeddings are computed by applying graph filtrations on positional encodings.

The proposed schema is analysed both theoretically and practically on different chemical datasets and one synthetic tree task.

**Strengths:**

The paper is well written and easy to follow.
Also, the idea of introducing topological aware positional encoding for graphs is interesting.

**Weaknesses:**

## Limited contribution
In my opinion, the only contribution of the paper is the computation of topological embeddings starting from positional encodings instead of node embeddings. The propagation of positional encodings and the computation of topological embeddings is not new. The contribution would be enough if the paper was able to convince why (analytically and practically) this could lead to better results.

## Meaning of theoretical results

It was not clear to me the purposes of the theoretical analysis of the paper. Proposition 1 shows the stability of the proposed method but it is not clear why we need this property. Lemma 1, Proposition 2 and Proposition 3 show the expressiveness of the proposal. In particular, it shows that ToPE is more expressive than standard PE. From my understanding, this is due to the employment of persistent homology features whose expressiveness has been already proven. I believe that the most interesting thing is to prove that the proposed method (i.e. combining positional and topological embeddings) is better than employing only a PH-based method (e.g RePHINE). However, there is not proof in this sense.

## Experimental Results and Reproducibility Issues

The experiments are limited since they have been conducted only on graph molecules and a synthetic tree manipulations. The abstract claims “on several graph and language processing applications, including molecular property prediction, out-of-distribution generalization, and synthetic tree tasks.” which is an overstatement.
Also, the results show only marginal improvements. If we consider the variance of the results, the difference might be non-statistically significant. As a side note, the tables with the results are a little bit confusing for me since the name of the proposed method (ToPE) never appears.

There is no baseline without positional encodings (e.g. RePHINE) to verify if the proposed method is more effective than these types of approaches.

Finally, there is no mention of “model selection” in the paper. To ensure reproducibility, it is not enough to publish the best hyperparameters set. Instead, it is necessary to publish the whole experimental procedure (i.e. the method for the model selection, the split used, the hyper-parameters grid, etc…) together with the code. Thus, I couldn't verify how the experiments have been conducted.

**Questions:**

I do not have specific questions more than the doubts expressed in the above section.

---

> ### Author Response · Authors · 2024-11-24
> **Response**
>
> Thank you for your feedback! We address all your questions and comments below.
>
> > In my opinion, the only contribution of the paper is the computation of topological embeddings starting from positional encodings instead of node embeddings.
>
> At a conceptual level, our contribution lies at the fact that by integrating PH-based topological descriptors on top of PE methods we can leverage multiscale global information not captured by prior PEs alone. This is the rationale why ToPE should lead to better results, and we have established the enhanced expressivity formally. Additionally, in response to Reviewer TQ3o's feedback, we have expanded our analysis to prove that ToPE is strictly more expressive than PEs when combined with GNNs.
>
> From an architectural stance, ToPE extends previous GNNs by intertwining graph structure, node features/embeddings, global positional information, and global topological features. In particular, TOGL and RePHINE generate filtrations using layer-wise GNN embeddings derived from input-attributed graphs, whereas ToPE relies solely on structural information captured through initial positional encodings. Additionally, TOGL and RePHINE do not utilize positional encodings. In contrast, ToPE integrates both topological and positional encodings as inputs to the backbone GNN.
>
> > It was not clear to me the purposes of the theoretical analysis of the paper.....
>
> Thanks for your comment.
> Since ToPE is a positional encoding approach, let us first consider unattributed input graphs. In this case, ToPE is more expressive than the RePHINE overall model. We note that RePHINE model is a particular case of ToPE --- we recover RePHINE from ToPE if we adopt constant vectors as positional encodings and do not input topological and positional embeddings to the backbone GNN. Then, to show that ToPE is strictly more expressive, it suffices to find two non-isomorphic graphs with the same number of connected components but different spectra (including eigenvectors) that 1-WL cannot distinguish. The graph representation (only carbon atoms) of the decalin and bicyclopentyl molecules provide such examples. The reason why ToPE can distinguish these is they have different spectra --- using LapPE will provide different filtration functions (and corresponding diagrams). On the other hand, since they are 1-WL indistinguishable and all nodes have the same color (carbon atom), then both the diagrams and the GNN embeddings will be identical at each layer.
>
> For attributed graphs, we can achieve similar result if we additionally incorporate GNN embeddings when computing persistence (RePHINE) diagrams --- this way ToPE will generalize RePHINE/TOGL.
>
> > The experiments are limited since they have been conducted only on graph molecules and a synthetic tree manipulations.
>
> We apologize for the statement regarding natural language processing applications. However, we respecfully disagree for graph experiments. We have conducted a rigoruous set of experiments to showcase the enhanced performance improvements described as below:
>
> - **Drug Property Prediction and Graph Classification**: We used the ZINC and Alchemy datasets which are widely used in drug discovery domain, with the main goal to predict molecular properties. For graph classification tasks, we employed OGBG-MOLTOX21 and OGBG-MOLPCBA datasets. Our findings (Fig. 3) **reveal 15-30 % increase in performance on drug property prediction, and **upto 15 % increase in accuracy for graph classification**.
> - **OOD Generalization**: To assess our method's ability to handle domain shifts, we utilized DrugOOD, an out-of-distribution (OOD) benchmark. As illustrated in Table 1, our approach **achieves the best OOD-Test accuracy among all domains**.
> - **Synthetic Tree Tasks**: We demonstrate the utility of our method in  synthetic tree-tasks involving binary branching tree. The results presented in Table 2, shows that **our method achieves the lowest PPL in 16 out of 18 tasks.**

---

> > ### Author Response · Authors · 2024-11-24
> > **Response Contd**
> >
> > >  There is no baseline without positional encodings (e.g. RePHINE) to verify if the proposed method is more effective than these types of approaches.
> >
> >
> > We have proposed a positional encoding method, and to ensure a fair comparison and demonstrate the benefit of our method, we have adhered to the same data preparation strategy and training/validation/testing splits and compared our method to latest graph positional encoding methods such as SPE[1], PEG[2], SignNet[3], LapPE and RWPE[4].
> >
> > [1] Yinan Huang, et. al, On the stability of expressive positional encodings for graph neural networks, ICLR 2024.
> >
> > [2] Haorui Wang,et. al, Equivariant and stable positional encoding for more powerful graph neural networks, ICLR 2022.
> >
> > [3] Derek Lim,  et. al, Sign and basis invariant networks for spectral graph representation learning, ICLR 2023
> >
> > [4] Vijay Prakash Dwivedi, et. al, Graph neural networks with learnable structural and positional representations, ICLR 2022
> >
> > > Finally, there is no mention of “model selection” in the paper.....
> >
> > We do not perform any hyper-parameter optimization. To ensure a fair comparison, we follow the same data preparation strategies, model architecture and training scheme as described in SPE[1].
> >
> > [1] Yinan Huang, et. al, On the stability of expressive positional encodings for graph neural networks, ICLR 2024.
> >
> > We are grateful for your review. We hope our response has addressed your questions and concerns.

---

### Official Review · Reviewer_TQ3o · 2024-11-03

**Soundness:** 2
**Presentation:** 3
**Contribution:** 2
**Rating:** 5
**Confidence:** 5

**Summary:**

In this article, the authors present a method to add some additional features, called Topological Positional Embeddings (ToPE), to vertices during GNN's update rule. The authors present their method, prove some properties of ToPE, and present extensive numerical experiments.

**Strengths:**

- The paper clearly exposes the contributions.

- Comparing the expressivity of different positional embeddings is relevant.

- The article has extensive experiments that could guide practitioners if they wish to implement topological positional encodings for GNNs.

**Weaknesses:**

I have two major concerns about the theoretical contributions. I am not sure this part of the article is very insightful from that standpoint. As a consequence, I am not entirely sure the methodological contribution is significant. My concern may be dissipated depending on the author's answers to my questions below.

**Questions:**

- The authors indeed show that ToPE has more expressive power than Laplacian PEs (individually); but it isn't clear that within aggregate-combine GNNs, the first is more expressive than the second (stricly). In other words, it could be true for any GNN with ToPE, there is a Laplacian PE GNN that can distinguish any pair of graphs that the TopE distinguishes. Can the authors elaborate on why persistent homology provides GNNs more expressivity than Laplacian PE?

- In Definition 1 of the Stable PE, the constant L_{\psi} depends on the graph given the order of the quantifiers. This constant could get very large, a priori, even if the graphs are close, and lead to unstability. The authors probably want to change the order of the statement. If not, could the authors elaborate on this observation / issue?

- The statement of Proposition 3 is confusing. Does the existential result apply to the hash function? Can the authors reformulate this statement in order to make the role of the Hash function clearer?

---

> ### Author Response · Authors · 2024-11-24
> **Response**
>
> Thanks so much for your thoughtful comments and excellent suggestions! We've acted on all of them, and also address all your concerns, as we describe below.
>
> > I have two major concerns about the theoretical contributions. I am not sure this part of the article is very insightful from that standpoint.
>
> From a theoretical perspective, we focus on establishing results to motivate the combination of PH-based topological descriptors and PEs. In this regard, we show that incorporating topological descriptors boosts the representational power of base PE methods. Thanks to your comments, **we have also extended our initial analysis to establish that ToPE is strictly more expressive than PEs combined with GNNs** (see answer below). **We have additionally shown that ToPE is more powerful than RePHINE (diagram + GNN)** (please, see answer to 7FDC). The complementary result regarding the connection to k-FWL (**Proposition 3**) aims to strengthen previous findings in the literature.
>
> > I am not entirely sure the methodological contribution is significant. My concern may be dissipated depending on the author's answers to my questions below.
>
> We first note that our paper is the _first work_ to propose enhancing positional encodings via PH-based topological descriptors. Our choice of descriptor leverages multiscale global information not captured by prior methods (e.g., LapPE). In addition, our approach extend previous GNNs by intertwining graph structure, node features/embeddings, global positional information, and global topological features. Finally, we also show the effectiveness on multiple benchmarks, including drug property prediction (**up 15-30% increase in performance**), graph classification (**up to 15\% increase in accuracy**), OOD generalization (**best among all domains, Table 1**), and synthetic tree tasks (**lowest PPL in 16 out 18 tasks**, Table 2).
>
> > The authors indeed show that ToPE has more expressive power than Laplacian PEs (individually); but it isn't clear that within aggregate-combine GNNs,....
>
> Thanks for you question. Indeed, ToPE is more expressive than LapPEs + aggregate-combine GNNs. We provide a proof sketch here.
>
> Let $C_{i}$ denote a cycle graph with $i$ nodes without node attributes. Consider a graph $G = \cup_{i=1}^{3n} C_{4}$, having $3n$ connected components and $12n$ nodes. Also, consider $G' = \cup_{i=1}^{n} C_{6} \cup C_{6}$, having $12n$ nodes and $2n$ connected components. Since, $G$ and $G'$ have the same number of nodes and identical local neighborhoods, aggregate-combine GNN cannot distinguish them. However, since the number of components in $G$ and $G'$ are different, they have different 0-dimensional persistence diagram $\mathcal{D}^{0}(G) \neq \mathcal{D}^{0}(G')$ for any filtration function. This difference in persistence diagrams allows ToPE to distinguish between the graphs. Note that, for LapPE methods with $k \leq 2n$, the $k$ first eigenvalues/eigenvectors of $G$ and $G'$ are identical and thus LapPE does not add any distinguishability power.
>
>
> > In Definition 1 of the Stable PE, the constant $L_{\psi}$...
>
> Thank you for bringing this to our attention. We wil fix it in the paper.
>
> > The statement of Proposition 3 is confusing. Does the existential result apply to the hash function?
>
> Thanks for your comment. By hash function we mean any injective function. The fact that the graphs are finite and the stable colors are natural numbers (i.e., the input to the hash function is a finite collection of natural numbers) implies the existence of such a function.
>
> Thank you so much for your constructive feedback. We hope we have sufficiently addressed your concerns.

---

> > ### Comment · Reviewer_TQ3o · 2024-11-24
> >
> > Thanks for your reply.
> > The proof you gave for ToPE vs LapPEs + agg-comb GNNs show that there are structures those LapPEs GNNs cannot distinguish that ToPE can. However the converse could also be true, and I am not seeing the proof of the second part of the statement of proposition 2 (in the appendix), as you simply mention ``To prove the Proposition 2, it suffices to i) show a pair of graphs that have same n smallest eigenvalue and eigenvector pairs, ii) show that persistence diagrams for those two graphs are different.''
> >
> > Could you please elaborate on this?

---

> > > ### Author Response · Authors · 2024-11-25
> > > **Response**
> > >
> > > Thanks for engaging in the discussion. In the following, we clarify the steps of the proofs.
> > >
> > > **Proof of Proposition 2 (original paper)**. Thanks for pointing this out. Indeed, in Proof of Proposition 2, we forgot to point to Lemma 1, which shows that persistent diagrams contain information from the positional encodings. In particular, let $Z$ and $Z'$ be the multiset of positional encodings (here treated as colors) for two graphs $G$ and $G'$. Then, the corresponding (0-dim) diagrams obtained from sublevel filtrations induced by the colors $Z$ and $Z'$ differ. This Lemma is a direct consequence (Corollary) of Lemma 5 in [1], which shows that the birth times of the diagrams contain these colors. We will refer to Lemma 1 in Proposition 2 in the revised manuscript.
> > >
> > > **The proof step in ToPE vs LapPE$+$GNN.**  One of the reasons why there are no structures that Lap+GNN can distinguish and ToPE cannot is that LapPE+GNN is a particular case of ToPE. Specifically, note that we input both the positional and topological embeddings to the backbone GNN. If we set the parameters of the GNNs to ignore the topological embeddings, we obtain an MP-GNN that incorporates layer-wise LapPE encodings.
> > >
> > > Even if you choose not to consider the backbone GNN in the analysis, the positional encoding information is still captured by the birth times of the persistence tuples and, consequently, their corresponding vectorizations (topological embeddings). Again, this is a direct consequence of the diagrams containing the filtration values (values) used to obtain the diagrams (Lemma 5 in [1]). This is also why Topological GNNs [2] are at least as expressive as GNNs (1-WL) -- e.g., see Theorem 3 in [2].
> > >
> > >
> > > [1] Going beyond persistent homology using persistent homology. NeurIPS, 2023.
> > >
> > > [2] Topological graph neural networks. ICLR, 2022.
> > >
> > > Thanks again for your careful reading. Please let us know if our answers are unclear and if we should provide further clarifications. Also, we will be happy to engage further if you have other concerns.

---

> > > > ### Comment · Reviewer_TQ3o · 2024-12-01
> > > >
> > > > Thanks for your detailed answer. I will maintain my score as is.

---

### Official Review · Reviewer_7fDC · 2024-11-04

**Soundness:** 3
**Presentation:** 2
**Contribution:** 2
**Rating:** 3
**Confidence:** 3

**Summary:**

ToPE computes positional encodings by learning them through Laplacian eigenvectors. The key difference is that ToPE employs persistent homology strategies, which are decoupled from the input graph and its features, and subsequently concatenates these encodings to the learned GNN representations.

**Strengths:**

- The experimental results are strong, with ToPE outperforming existing methods on most tasks.
- The paper is generally well-written.

**Weaknesses:**

- The novelty of the approach is unclear.
- The comparison to related work is not well articulated.
- The advantages of ToPE over REPHINE and other PH-based methods are not clearly demonstrated/explained.
- Proposition 1 is trivial, and the other results do not contribute significant new insights because
a) They do not analyze the actual final GNN, and
b) The proofs are overly straightforward.

**Questions:**

- What is the novelty wrt to related work such as rephine? Is it just using Laplacian eigenvectors as input instead of the initial node features?
- While you analyze the stability and expressivity, can you also compare it with other methods? E.g., is it "more stable" than SPE? Is it more expressive than rephine when using the corresponding topological descriptor, i.e., using rephine diagrams?


---
## Additional Comments
I believe the paper isn't ready for publication, yet. I would recommend embedding it better into the literature, pointing out the novelties, compared to other works. In particular, the authors should point out the tasks on which they suggest using Tope and when they expect it to perform better/worse than other methods.

---

> ### Author Response · Authors · 2024-11-24
> **Response**
>
> Many thanks for your constructive review. We address your concerns and incorporate your suggestions below.
>
> > The novelty of the approach is unclear.
>
> In sum, the novelty our paper mainly stems from the fact that it is the _first work_ to enhance positional encodings with PH-based topological descriptors. Notably, these descriptors provide multiscale global information not captured by prior methods.
>
> > The comparison to related work is not well articulated [...] What is the novelty wrt to related work such as rephine? Is it just using Laplacian eigenvectors as input instead of the initial node features?
>
> From an architectural viewpoint, the differences to RePHINE are:
> - No positional encodings are used in RePHINE;
> - RePHINE embeddings are not input to the backbone GNN, whereas we propose integrating both layer-wise positional embeddings, topological vectors into node-level updates of the GNN.
> - Our architecture uses two GNNs while RePHINE uses one;
> - As you correctly pointed out, since we propose using PH-based topological descriptors as positional encodings, our filtration functions are only defined based on structural information whereas RePHINE use node embeddings (including initial node features).
>
> In addition, as far as we know, our paper is the _first work_ to enhance positional encodings with PH-based topological descriptors. Importantly, these descriptors provide multiscale global information not captured by prior methods (e.g., LapPE).
>
> > Proposition 1 is trivial, and the other results do not contribute significant new insights because a) They do not analyze the actual final GNN, and b) The proofs are overly straightforward.
>
> Thanks for your comment. We agree that our stability result does not apply to the full model (including the diagram). Thus, we will move it to the Appendix.
>
> > The advantages of ToPE over REPHINE and other PH-based methods are not clearly demonstrated/explained...
>
> This is a great question.
>
> Since ToPE is a positional encoding approach, let us first consider unattributed input graphs. In this case, ToPE is more expressive than the RePHINE overall model (diagrams + GNN). We note that RePHINE model is a particular case of ToPE --- we recover RePHINE from ToPE if we adopt constant vectors as positional encodings and do not input topological and positional embeddings to the backbone GNN. Then, to show that ToPE is strictly more expressive, it suffices to find two non-isomorphic graphs with the same number of connected components but different spectra (including eigenvectors) that 1-WL cannot distinguish. The graph representation (only carbon atoms) of the decalin and bicyclopentyl molecules provide such examples. The reason why ToPE can distinguish these is they have different spectra --- using LapPE will provide different filtration functions (and corresponding diagrams). On the other hand, since they are 1-WL indistinguishable and all nodes have the same color (carbon atom), then both the diagrams and the GNN embeddings will be identical at each layer.
>
> For attributed graphs, we can achieve similar results if we incorporate GNN embeddings when computing persistence diagrams.
>
> We thank you for your constructive feedback, which has strengthened our work. We hope our answers have alleviated some of your concerns.

---

> > ### Comment · Reviewer_7fDC · 2024-11-26
> >
> > Dear Authors,
> >
> > Thank you for your reply. I have a few points and questions for clarification:
> >
> > 1) Could you elaborate on the specific "multiscale global information" your approach captures that standard LPEs or Rephine cannot?
> >
> > 2) I believe it’s important to emphasize that your methodology combines Laplacian eigenvectors, Rephine, and GNNs, applying them sequentially to input graphs. This aspect is not made sufficiently clear in your manuscript. It is entirely valid to concatenate prior approaches, especially if doing so offers theoretical or experimental advantages, but this should be explicitly stated/made clearer in the manuscript.
> >
> > 3) I remain unclear on the expressivity advantage of your method. You provide an example where eigenvectors can distinguish certain structures that message passing cannot. However, there are filtrations that can separate decalin and bicyclopentyl. Moreover, Rephine alone should theoretically distinguish these structures without requiring positional encodings. Please correct me if I am mistaken.
> > As a result, I am uncertain whether combining Laplacian PEs, Rephine, and GNNs offers a tangible expressivity advantage compared to using any two of these components (e.g., LPEs+GNNs, LPEs+Rephine, or Rephine+GNNs) or even just Rephine alone.
> >
> > I would appreciate your clarification on these points.
> >
> > Best regards,

---

> ### Author Response · Authors · 2024-11-27
> **Response**
>
> Thank you for replying and engaging in the discussion. We address your concerns below:
>
> > Could you elaborate on the specific “multiscale global information” your approach captures that standard LPEs or Rephine cannot?
>
> By multiscale global information, we mean the property of PH-based methods to account for topological invariants at different resolutions of input data (given by the filtration steps). In our case, global information refers to connectedness as we use 0-dim PH (or RePHINE). When dealing with unattributed graphs, as with any positional encoding, RePHINE (or any color-based PH method) is very limited as all nodes share the same colors. On the other hand, due to the inherent truncated aspect of LapPE, it may fail to capture connectedness (as we show in Proposition 2). Therefore, using LapPE to define filtration function enhances the expressivity of these approaches alone.
>
> > I believe it’s important to emphasize that your methodology combines Laplacian eigenvectors, Rephine, and GNNs, applying them sequentially to input graphs. This aspect is not made sufficiently clear in your manuscript. It is entirely valid to concatenate prior approaches, especially if doing so offers theoretical or experimental advantages, but this should be explicitly stated/made clearer in the manuscript.
>
> Thanks for your suggestion. The integration of the topological + positional embeddings into the GNN is described in Eq. (3) and Figure 1. However, as you suggested, we will revise the paper to make it clearer.
>
> >  I remain unclear on the expressivity advantage of your method. You provide an example where eigenvectors can distinguish certain structures that message passing cannot. However, there are filtrations that can separate decalin and bicyclopentyl. Moreover, Rephine alone should theoretically distinguish these structures without requiring positional encodings. Please correct me if I am mistaken.
>
> In the RePHINE original paper, the authors consider filtrations induced by node features/colors. When all colors are the same, RePHINE and vertex-color (VC) filtrations can only capture the number of components --- i.e., the tuples are either trivial (same birth and death times) or $(\cdot, \infty)$. Thus, VC-based PH or RePHINE cannot distinguish decalin and bicyclopentyl without PEs --- both decalin and bicyclopentyl contains one connected component, hence, filtrations defined on GNN node embeddings or initial node features (without positional encodings) will result in the same 0-dim persistence diagram and cannot be separated via PH or PH+GNN.
>
> You are right when you say that "there are filtrations that can separate decalin and bicyclopentyl". In fact, we propose using PEs (LapPEs) to define such filtrations. The central idea of our contribution is to use PEs and their embeddings after each layer of a GNN to obtain layer-wise topological embeddings via persistent homology. These topological embeddings serve as novel positional encodings

---

> > ### Comment · Reviewer_7fDC · 2024-11-29
> >
> > Thank you for replying
> >
> > Could you please answer my last question from above: Let us consider a well-defined LPE method, such as SPE from Huang et al. or BasisNet from Lim et al., and assume a) the use of the complete eigendecomposition, and b) the use of partial eigendecomposition.
> > Is there any expressivity advantage of $\mathrm{GNN} \circ\mathrm{RePHINE} \circ \mathrm{LPE}$ over $\mathrm{RePHINE} \circ \mathrm{LPE}$ or $\mathrm{GNN} \circ\mathrm{LPE}$?
> >
> > Best regards,

---

> ### Author Response · Authors · 2024-12-03
> **Response**
>
> Thanks for your question. We reflected on it and this is our response:
>
> Although one could consider expressivity wrt graphs with number of nodes smaller than k (dimensionality of positional encodings), this is not a standard setting. To the best of our knowledge, no positional encodings rely on full eigendecomposition.
>
> Therefore, in the following, we only consider settings where the dimensionality of the PE method, k, is smaller than the number of nodes, n --- i.e., assumption (b) in our comment:
>
> **GNN $\circ$ RePHINE $\circ$ LPE is more expressive than RePHINE $\circ$ LPE**
>
> Let $C_{i}$ denote a cycle graph with $i$ nodes without node attributes, and $P_{i}$ denote a path graph with $i$ nodes without node attributes. Consider graphs $G = \cup_{i=1}^{n}C_{4}$ and $G’ = \cup_{i=1}^{n}P_{4}$, which have $n$ connected components and $4n$ nodes.  The $k$ smallest laplacian eigenvalues corresponding to $G$ and $G’$ are all equal to 0 with the identical constant eigenvector, where $k < 4n$, and both graphs have the same number of connected components. Hence, LPE relying on partial decomposition and RePHINE relying on 0-dim persistence diagrams to determine the number of components would not be able to separate these two graphs. However, the nodes in $G$ and $G’$ have different degrees, allowing GNNs (in ToPE) to distinguish these two graphs. Finally, note that RePHINE $\circ$ LPE is a particular case of GNN $\circ$ RePHINE $\circ$, thus the latter cannot be less expressive.
>
> **GNN $\circ$ RePHINE $\circ$ LPE is more expressive than GNN $\circ$ LPE**
>
> Let $C_{i}$ denote a cycle graph with $i$ nodes without node attributes. Consider a graph $G = \cup_{i=1}^{3n} C_{4}$, having $3n$ connected components and $12n$ nodes. Also, consider $G’ = \cup_{i=1}^{n} C_{6} \cup C_{6}$, having $12n$ nodes and $2n$ connected components. Since, $G$ and $G’$ have the same number of nodes and identical local neighborhoods, aggregate-combine GNN cannot distinguish them. However, since the number of components in $G$ and $G’$ are different, they have different 0-dimensional persistence diagram $\mathcal{D}^{0}(G) \neq \mathcal{D}^{0}(G’)$ for any filtration function. This difference in persistence diagrams allows ToPE to distinguish between the graphs. Note that, for LapPE methods with $k \leq 2n$, the $k$ first eigenvalues/eigenvectors of $G$ and $G’$ are identical and thus LapPE does not add any distinguishability power. Again, since the LPE colors are captured in RePHINE diagrams, GNN $\circ$ RePHINE $\circ$ LPE is not less expressive.
>
> Thank you again for engaging with us.

---

### Official Review · Reviewer_scmW · 2024-11-06

**Soundness:** 2
**Presentation:** 2
**Contribution:** 1
**Rating:** 3
**Confidence:** 3

**Summary:**

This paper proposes to integrate persistent homology (PH), combined with positional encodings, into GNNs to boost expressivity. Specifically, topological positional encoding (ToPE) is proposed, which uses PEs to induce graph filtration and the obtained PH's embeddings are concat with base PEs and fed into GNNs.

**Strengths:**

1. The method is straightforward.
2. The manuscript is easy to follow.

**Weaknesses:**

1. Related work is not contextualized enough, making the original contributions of this work not well presented. What are the key contributions of this work compared with previous work? It remains unclear to me what is the key innovation of ToPE compared with VC and RePHINE.
2. It is good to contain some theoretical analysis, but it may not be a core contribution of this work as they are somewhat shallow and artificial.
3. The proposed way of combining PEs and PH seems arbitrary and with limited technical novelty, making it sound more like an engineering trick instead of a novel and principled method.

**Questions:**

See above.

---

> ### Author Response · Authors · 2024-11-24
> **Response**
>
> Many thanks for your feedback and suggestions. We address all your comments below.
>
> > Related work is not contextualized enough......
>
> Thank you for the opportunity to elaborate on our contributions.
>
> As far as we know, our paper is the _first work_ to enhance positional encodings with PH-based topological descriptors. Importantly, these descriptors provide multiscale global information not captured by prior methods (e.g., LapPE).
>
> **Methodological difference to VC/RePHINE**: Regarding the difference to TOGL/RePHINE, we note that TOGL/RePHINE uses the layer-wise GNN embeddings from input attributed graphs to obtain filtrations, whereas ToPE only leverages structural information through initial positional encodings. No positional encodings are used in RePHINE/TOGL. Unlike TOGL/RePHINE, in ToPE, both the topological and positional encodings are fed to the backbone GNN.
>
> **Theoretical Analysis** The focus of our theoretical analysis is to show that incorporating topological descriptors boosts the representational power of base PE methods. In that sense, our theoretical analysis demonstrates ToPE's superior expressivity over simple PE methods (**Proposition 2, Lemma 1**). Motivated by comments from Reviewer TQ3o and , \textbf{we have extended our initial analysis to establish that ToPE is also strictly more expressive than PEs combined with GNNs}. Moreover, thanks to questions by reviewers 7fDC and 72Z5, **we have additionally shown that ToPE is more expressive than RePHINE (diagram + GNN)**. Finally, we also establish ToPE's connection to k-FWL (**Proposition 3**).
>
> **Empirical results.** We showcase the effectiveness of ToPE by performing a rigorous set of experiments:
> - **Drug Property Prediction and Graph Classification**: We used the ZINC and Alchemy datasets, with the main goal to predict molecular properties. For graph classification tasks, we employed OGBG-MOLTOX21 and OGBG-MOLPCBA datasets. Our findings (Fig. 3) **reveal $15-30 \%$ increase in performance on drug property prediction, and upto $15\%$ increase in accuracy for graph classification**.
> - **OOD Generalization**: To assess our method's ability to handle domain shifts, we utilized DrugOOD, an out-of-distribution (OOD) benchmark. As illustrated in Table 1, our approach **achieves the best OOD-Test accuracy among all domains**.
> - **Synthetic Tree Tasks**: We demonstrate the utility of our method in  synthetic tree-tasks involving binary branching tree. The results presented in Table 2, shows that **our method achieves the lowest PPL in 16 out of 18 tasks.**
>
> > It is good to contain some theoretical analysis, but it may not be a core contribution of this work as they are somewhat shallow and artificial.
>
> We agree that our theoretical analysis may not be the core of our work. Nonetheless, importantly, it demonstrates the expressivity gains of our approach, which was our initial motivation. As mentioned in our previous answer, we have extended our analysis to show that ToPE is more expressive than the combination of PEs + GNNs.
>
> > The proposed way of combining PEs and PH seems arbitrary and with limited technical novelty, making it sound more like an engineering trick instead of a novel and principled method.
>
> We respectfully disagree. Our method is grounded on the idea that PH-based topological descriptors provide complementary important information not captured by existing PE methods. Also, our proposal is rather flexible as it comprises general message-passing GNNs, PEs, and PH-based descriptors. Therefore, it corresponds more to a general principle than a combination of specific methods.
>
> We are grateful for your comments and suggestions, which have allowed us to emphasize some salient aspects, and shed light on subtle facets of the proposed method.

---

### Meta-Review · Area_Chair_cRCN · 2024-12-20

**Metareview:**

This submission proposes the integration of persistent homology (PH) with positional encodings into Graph Neural Networks (GNNs), introducing Topological Positional Encoding (ToPE). ToPE leverages positional encodings to induce graph filtration, combining the resulting PH embeddings with base positional encodings as input to GNNs. While the idea is intriguing, the paper unfortunately has a fundamental misconception regarding the relationship between structural representations and positional embeddings, a topic well-established in the literature.

Notably, even the abstract contains an incorrect statement claiming that equivariant representations are incapable of encoding global structure, which is provably false. These and other errors in the paper undermine the foundation of the paper and raises concerns about the soundness of the approach. Furthermore, the proposed use of learnable persistence diagrams to incorporate topological features into positional encodings does not constitute a substantial contribution, as it builds upon existing ideas without providing significant novel insights.

To strengthen this work, I recommend that the authors conduct a more thorough examination of the theoretical underpinnings of positional and structural representations. A more meaningful contribution could be made by revisiting the fundamental concepts, identifying knowledge gaps, and propose solutions that address these gaps. While the paper shows promise, its current version requires significant revisions to demonstrate a more substantial contribution. As such, I do not recommend acceptance in its present form.

**Additional Comments On Reviewer Discussion:**

Discussion was not meaningful. The paper needs too many corrections for a rebuttal.

---

### Decision · Program_Chairs · 2025-01-22

Reject